# A Method of Infrared Small Target Detection in Strong Wind Wave Backlight Conditions

**Dongdong Ma, Lili Dong * and Wenhai Xu**

School of Information Science and Technology, Dalian Maritime University, Dalian 116026, China; dlmu0120200060@dlmu.edu.cn (D.M.); wenhaixu@dlmu.edu.cn (W.X.)
* Correspondence: donglili@dlmu.edu.cn; Tel.: +86-138-8968-1193

**Abstract:** How to accurately detect small targets from the complex maritime environment has been a bottleneck problem. The strong wind-wave backlight conditions (SWWBC) is the most common situation in the process of distress target detection. In order to solve this problem, the main contribution of this paper is to propose a small target detection method suitable for SWWBC. First of all, for the purpose of suppressing the gray value of the background, it is analyzed that some minimum points with the lowest gray value tend to gather in the interior of the small target. As the distance from the extreme point increases, the gray value of the pixel in all directions also increases by the same extent. Therefore, an inverse Gaussian difference (IGD) preprocessing method similar to the distribution of the target pixel value is proposed to suppress the uniform sea wave and intensity of the sky background. So as to achieve the purpose of background suppression. Secondly, according to the feature that the small target tends to "ellipse shape" in both horizontal and vertical directions, a multi-scale and multi-directional Gabor filter is applied to filter out interference without "ellipse shape". Combined with the inter-scale difference (IsD) operation and iterative normalization operator to process the results of the same direction under different scales, it can further suppress the noise interference, highlight the significance of the target, and fuse the processing results to enrich the target information. Then, according to different texture feature distributions of the target and noise in the multi-scale feature fusion results, a cross-correlation (CC) algorithm is proposed to eliminate noise. Finally, according to the dispersion of the number of extreme points and the significance of the intensity of the small target compared with the sea wave and sky noise, a new peak significance remeasurement method is proposed to highlight the intensity of the target and combined with a binary method to achieve accurate target segmentation. In order to better evaluate the performance index of the proposed method, compared with current state-of-art maritime target detection technologies. The experimental results of multiple image sequence sets confirm that the proposed method has higher accuracy, lower false alarm rate, lower complexity, and higher stability.

**Keywords:** maritime strong wind-wave backlight condition; infrared maritime target detection; multi-scale feature extraction; cross-correlation theory; peak significance remeasurement

## 1. Introduction

Nowadays, the rapid development and high reliability of infrared search and tracking systems has become an increasingly urgent need in the field of maritime search and rescue [1–3]. How to accurately detect small targets in complex sea conditions has been the essential issue of maritime search and rescue. The main work of this paper is to solve the problem of small target detection under SWWBC. At present, detection methods for small targets in the complex maritime environment are emerging one after another.

In the research based on contrast and similarity characteristics, C. L. Philip [4] based on the robustness of HVS, proposed a multi-scale local contrast measurement method based on the derived kernel model (DK Model) to achieve infrared target detection. Li [5] proposed a local adaptive contrast detection method (LACM-LSK) based on local steering

kernel reconstruction. Due to the improved local contrast method requirement that the target has a big difference in the local area compared with the clutter interference, the application scenario has greater limitations.

In the research based on texture directional features, aghaziyarati [6] proposed a method based on the cumulative directional derivative weighting coefficient to overcome the shortcomings of the average absolute gray difference (AAGD) algorithm. Moradi [7] constructed a new directional small target detection algorithm, called absolute directional mean difference (ADMD), using a concept similar to the average absolute gray difference. Wei [8] et al. decomposed the multi-scale image into horizontal direction by a wavelet transform method and define a "mutual wavelet energy composition" method (MWEC) to detect small infrared targets in the sea sky environment. Due to the weak ability of the above methods to suppress strong sea wave clutter, the false alarm rate will be higher when applied in SWWBC.

In the research based on statistical characteristics, Zhu [9] considered the inherent spatial correlation between image pixels to indicate that the background is continuous and highly correlated. On the contrary, the target is regarded as destroying the local correlation. Therefore, segmenting the target from the background can be regarded as the restoration of the low-rank matrix. Zhang [10] adopted a new non-convex low-rank constraint based on the infrared patch tensor (IPT) model, that is, the partial sum of tensor nuclear norm (PSTNN) joint weighted l1 norm to effectively suppress background. Due to the large amount of calculation in the above methods, real-time and practicality cannot be guaranteed.

In the research based on spatiotemporal characteristics, Zhao [11] proposed a small infrared moving target detection algorithm based on the spatiotemporal consistency of motion trajectory. Chiman [12] combined the optical flow method with contrast enhancement, connected component analysis, target association, and other methods to effectively perform target detection. The above method needs to calculate the corresponding relationship of feature points between different images, the detection platform must have the ability of antivibration, otherwise, the robustness to the complex noise environment is extremely poor.

In the research based on deep learning, Li [13] applied the deep learning maritime target detection model provided by Google, combined with the super-pixel segmentation algorithm to optimize the Grabcut algorithm, and discovered the deep learning marine target detection and segmentation, which can accurately extract the target contour and semantic information. Ryu [14] proposed a new far-infrared small target detection method based on deep learning and a heterogeneous data fusion method to solve the problem of the lack of semantic information due to the small target size. If the limited training samples contain not enough information, the target will not be well recognized in the deep learning method.

Recently, it has been noticed that the ability of the human visual attention system to detect objects from complex scenes of optical images is faster and more reliable [15,16]. Many excellent computational visual attention models have been proposed to simulate the structure of the human visual system.

Itti [17,18] proposed a visual attention system based on the behavior and neuron structure of the early primate visual system. First, Gaussian filter and Gabor filter and linear "center-surround" difference operation are applied to extract early visual features, then multi-scale image features under the same feature are linearly superimposed, and then the images under different features are linearly fused to form a single saliency map. Finally, Koch [19] proposed to filter the target location according to the dynamic neural network (WTA and IOR) in order from strong to weak. Dong [20] proposed a method based on the visual attention and pipeline-filtering model (VAPFM), the overall method adopts single-frame suspected target detection based on the improved visual attention model and multi-frame real target judgment based on anti-jitter (VAPFM). Wang [21] proposed a robust anti-jitter spatiotemporal saliency generation with parallel binarization

(ASSGPB) method. Using the spatial saliency and time consistency of the target, the real target is separated from the cluttered area. The above-mentioned target detection method is effective in some simple scenes, if the task is to detect targets in complex scenes, the above method may lose its effect such as in SWWBC. In view of the above problems, the proposed method should have the following properties:

(1) Lower false alarm rate; (2) Lower time-consumption; (3) Higher detection rate; (4) Higher stability.

In order to realize the above four attributes, this paper proposes a small target detection method in SWWBC. The final experimental results show that the proposed method is superior to the traditional and the latest target detection methods in detection performance. The rest of this paper is organized as follows. In Section 2, a small target detection method in SWWBC is introduced. Section 3 introduces the details of the experiment and analyzes the results. Section 4 summarizes the conclusion.

## 2. Materials and Methods

Figure 1 is the flow chart of our proposed structure for target detection, which is mainly divided into five steps, namely background suppression, feature extraction, noise elimination, target enhancement, and target segmentation. In the first step, the acquired infrared image is processed by IDG filter to generate the preprocessed image, the purpose is to suppress background intensity. In the second step, the preprocessed image is divided into two signal streams, which are processed by horizontal and vertical Gabor filters [22,23] in multi-scale, respectively, to generate multi-scale feature images. The multi-scale feature map of each direction is generated after the IsD operation and iterative normalization operator. The multi-scale feature map is fused and iterative normalization operation to generate saliency map, so as to eliminate noise and extract texture features. The third step is to generate a result map by CC calculation to filter the noise. In the fourth step, the local maxima in eight directions are obtained from the CC result image, in order to find the subsequent target segmentation points, the peak significance is re-measurement to achieve the target enhancement. Finally, using the binary segmentation method to determine the true target.

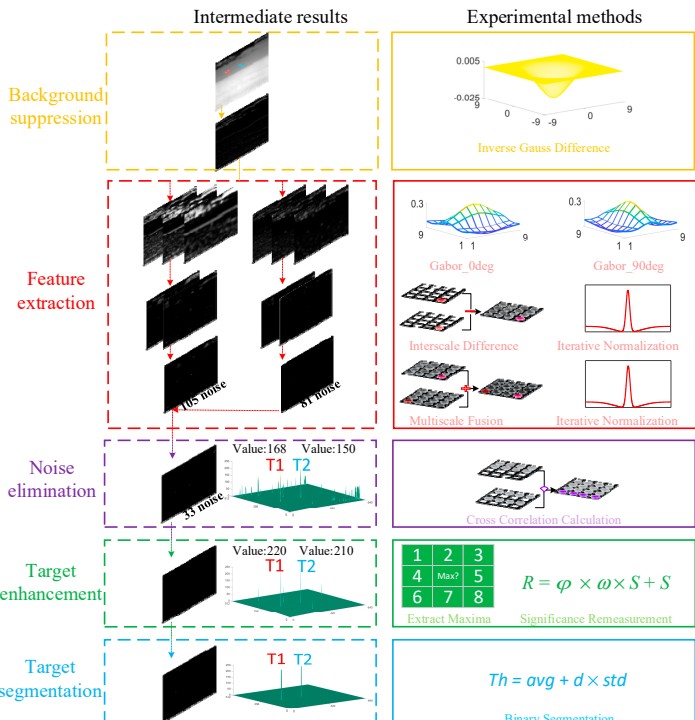

**Figure 1.** Structure flow chart of target detection method for SWWBC.

## 2.1. Background Suppression

Reducing the background gray value as much as possible is a key purpose of background suppression methods. In order to achieve the best background suppression effect, the analysis of the characteristics of the target and background is an essential process. Therefore, our paper first analyzes the weak small target patch and background patch in the typical infrared image, and the analysis results are shown in Figure 2. The background patch of Figure 2a is selected at the sea wave with the typical gray distribution. In Figure 2b, it is selected at the thick clouds and sea waves. It is found from the gray value distribution of the weak small target patch in (a) T1 and (b) T1 T2, the interior of the small target tends to gather some gray minimum points which are quite different from the surrounding pixels. With the increase of the distance from the minimum point, the gray values of the pixels in all directions raise by the same extent. However, the gray value distribution of (a) B1 B2 B3 sea wave patch and (b) B1 clouds patch and B2 sea wave patch has no similarity with the target patch.

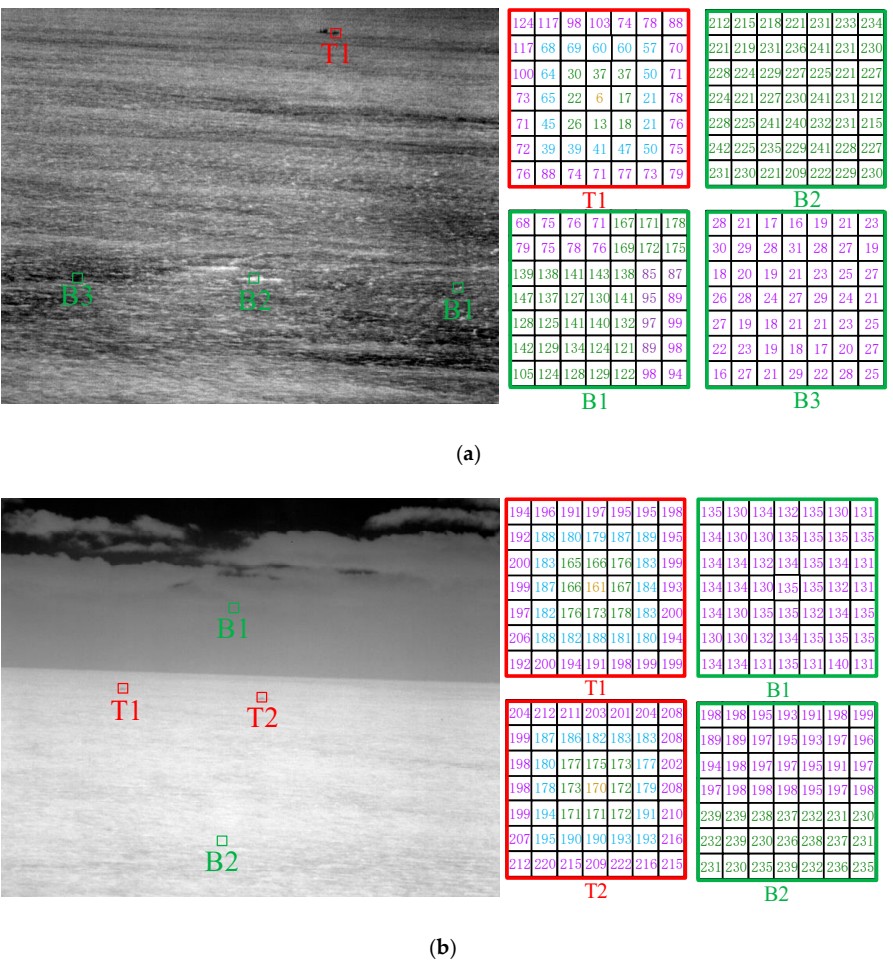

**Figure 2.** Analysis results of infrared small target block region and background patch region. (**a**) Typical image A infrared small target patch area and background patch area. (**b**) Typical image B infrared small target patch area and background patch area.

An IGD preprocessing algorithm similar to the target area gray value distribution is designed, the IGD preprocessing method is shown in Equation (1), $\sigma_1$ and $\sigma_2$ indicates the high-scale and low-scale filter parameters.

$$IGF(x,y) = \frac{1}{2\pi\sigma_1{}^2}e^{-\frac{x^2+y^2}{2\sigma_1{}^2}} - \frac{1}{2\pi\sigma_2{}^2}e^{-\frac{x^2+y^2}{2\sigma_2{}^2}} \tag{1}$$

The kernel function three-dimensional results are shown in Figure 3. From them, it can be found that the three-dimensional distribution of grayscale values in the target area is similar to the three-dimensional result of the kernel function, so the significance of the background can be better suppressed. According to the target space area in the statistical dataset is less than 80 pixels, IGD kernel size is selected as $9 \times 9$ in this paper. So that the spatial area of the kernel basically matches the spatial area of the target and the best background suppression effect can be obtained.

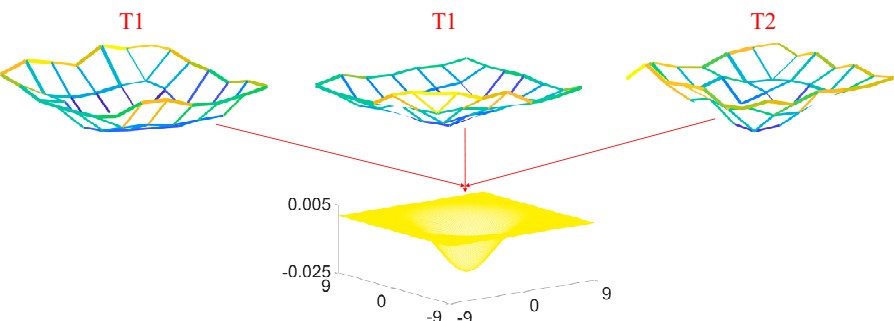

**Figure 3.** IGD kernel function.

The final obtained background suppression results are shown in Figure 4. Firstly, it can be seen by comparing the global gray histogram of the original image of typical images A and B with the global gray histogram of the processing image, gray values of the background after processing are decreased and substantially smaller than those of the target.

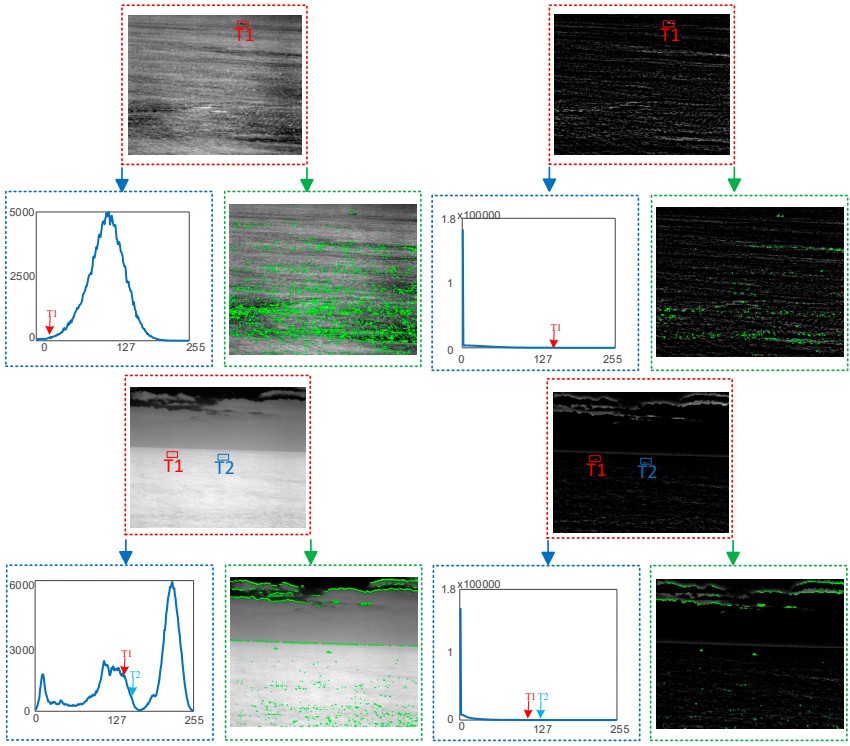

**Figure 4.** Global gray histogram and local contrast of typical original image and background suppression result.

Secondly, the local contrast is calculated by Equation (2), $g_f$ represents the average gray level of the foreground, and $g_b$ represents the average gray level of the background.

$$Contrast = \frac{max\left[g_f, g_b\right]}{min\left[g_f, g_b\right]} \tag{2}$$

For ease of observation, the competing areas with the same or higher local contrast as the target area are marked in green. The results show that the background interference area that has the same or higher local contrast as the target area after processing is significantly reduced, the reason for this result is also due to the reduction of the gray value of most backgrounds. Finally, it can also be found from the processed result image that the uniform sea waves and sky background in the original image are eliminated, which is also more beneficial to subsequent target detection tasks.

### 2.2. Feature Extraction

Accurate feature extraction [24,25] can better retain target information and eliminate interference information. By observing the resulting image after the background suppression, it can be found that the small target is closer to the "ellipse shape" and has strong texture characteristics in both the horizontal and vertical directions, and the Gabor filter just has the ability to extract the "ellipse shape" characteristics. In addition, the Gabor filter also has the function of multi-scale resolution [26,27]. By adjusting the size of the filter template to achieve "fine to coarse" feature extraction, we can ensure that the target region can be accurately extracted features. The mathematical expression of the Gabor filter is a deep representation in Equations (3) and (4). $\theta$ represent the filtering direction, $\gamma$ represent the aspect ratio, $\delta$ represent the standard deviation, $\lambda$ represent the wavelength, and $\psi$ represent the phase. Figure 5 shows the result of feature extraction of background suppression images A and B.

$$G(\theta, \gamma, \delta, \lambda, \psi, x, y) = \exp\left(-\frac{x'^2 + \gamma y'^2}{2\delta^2}\right) \cos\left(2\pi\frac{x'}{\lambda} + \psi\right) \tag{3}$$

$$\begin{cases} x' = x\cos(\theta) + y\sin(\theta) \\ y' = -x\sin(\theta) + y\cos(\theta) \end{cases} \tag{4}$$

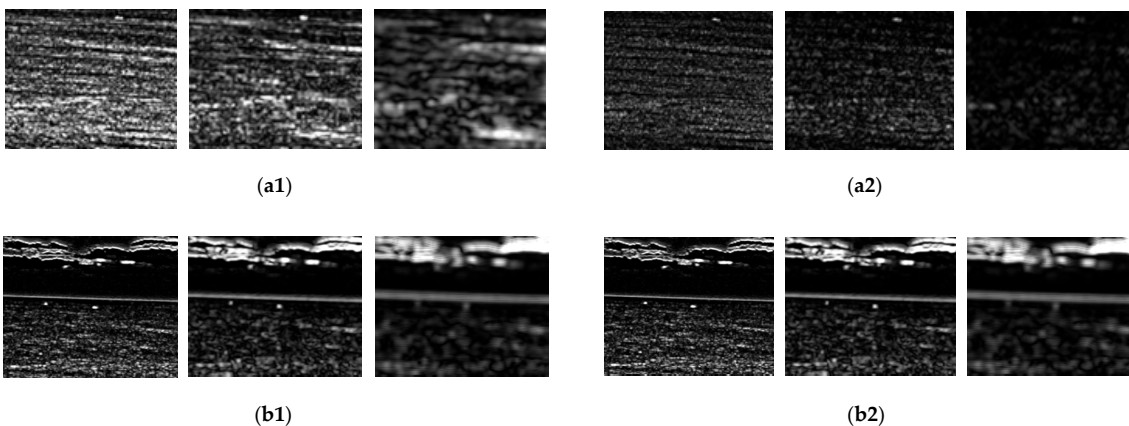

**Figure 5.** Multi-scale horizontal and vertical texture feature extraction results. (**a1**) Horizontal texture features in multi-scale image A. (**a2**) Vertical texture features in multi-scale image A. (**b1**) Horizontal texture features in multi-scale image B. (**b2**) Vertical texture features in multi-scale image B.

In the results of multi-scale horizontal and vertical texture feature extraction, it can be observed that the brightness of the target between adjacent scales in the same direction

gradually decreases, and the brightness of the sea wave and clouds between adjacent scales in the same direction gradually increases or remains unchanged. According to the intensity variation characteristics of targets, waves and clouds, subtract the results of different scales in the same direction, which is called "IsD operation". In the final "IsD operation" result, the position with a large difference is more likely to be the target, and the position with a small difference or negative difference is more likely to be the noise. Firstly, the negative difference is regarded as real noise and the value of the corresponding position is set to zero. Secondly, the iterative normalization operator is used to increase the intensity difference between the target and the noise and reduce the intensity of the noise to a negative number, so as to distinguish the target and the noise. The iterative normalization operator is shown in Equation (5), $c_{ex}$ and $\sigma_{ex}$ is stimulus factor and stimulus variance, respectively, which is used to further enhance the global highly significant region. $c_{inh}$ and $\sigma_{inh}$ is the suppression factor and the suppression variance respectively, which is used to further attenuate the global weak significance region. The results of multi-scale horizontal and vertical texture feature image processing are shown in Figure 6.

$$INorm(c_{ex}, \sigma_{ex}, c_{inh}, \sigma_{inh}, x, y) = \frac{c_{ex}^2}{2\pi\sigma_{ex}^2}e^{-(x^2+y^2)/2\sigma_{ex}^2} - \frac{c_{inh}^2}{2\pi\sigma_{inh}^2}e^{-(x^2+y^2)/2\sigma_{inh}^2} \qquad (5)$$

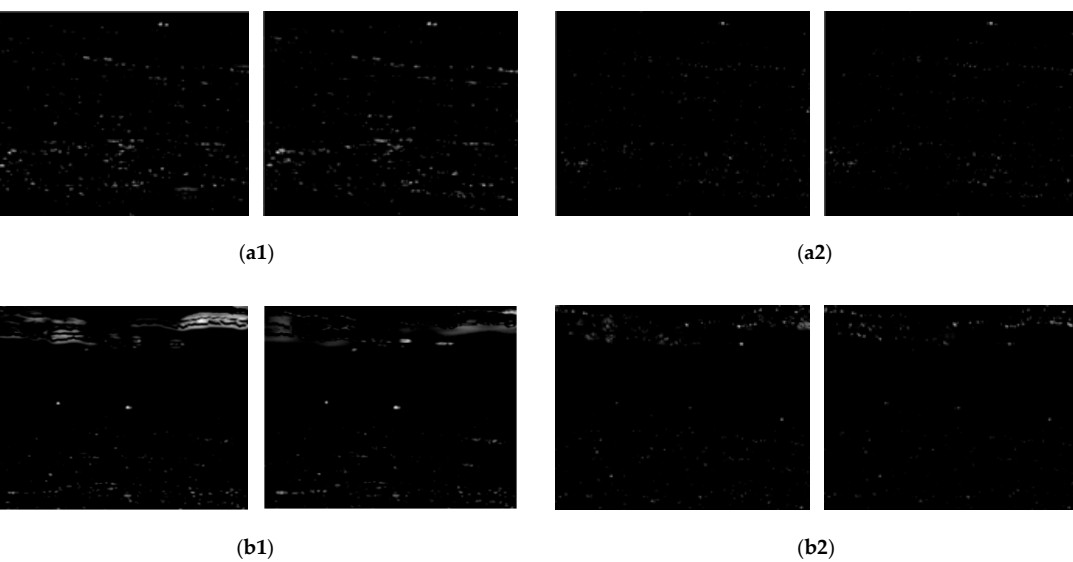

**Figure 6.** IsD operation results between horizontal and vertical scales. (**a1**) Inter-scale horizontal texture feature of image A. (**a2**) Inter-scale vertical texture feature of image A. (**b1**) Inter-scale horizontal texture feature of image B. (**b2**) Inter-scale vertical texture feature of image B.

In order to extract as much information of the target as possible, the IsD operation results with the same direction are fused, and the final fusion result is shown in Figure 7.

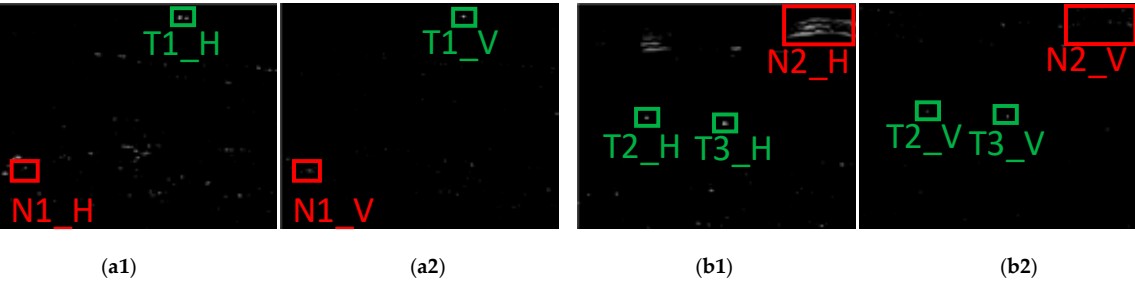

**Figure 7.** IsD Multi-scale feature fusion results (Algorithm 1). (**a1**) Image A horizontal fusion. (**a2**) Image A vertical fusion. (**b1**) Image B horizontal fusion. (**b2**) Image B vertical fusion.

---
**Algorithm 1:** Get the result of multi-scale feature fusion.

---
**Input:**
Gaussian difference preprocessing results *Gimg*.
**Output:**
Multiscale fusion results *MFR*.
1 : $G_{01} =$ Gimg $\otimes$ Gabor(0, 0.5, 2.3333, 7, 0, 9) Equation (3)
2 : $G_{02} = G_{01} \otimes$ Gabor(0, 0.5, 2.3333, 15, 0, 9) Equation (3)
3 : $G_{03} = G_{02} \otimes$ Gabor(0, 0.5, 2.3333, 21, 0, 9) Equation (3)
4 : $G_{901} =$ Gimg $\otimes$ Gabor(90, 0.5, 2.3333, 7, 0, 9) Equation (3)
5 : $G_{902} = G_{901} \otimes$ Gabor(90, 0.5, 2.3333, 15, 0, 9) Equation (3)
6 : $G_{903} = G_{902} \otimes$ Gabor(90, 0.5, 2.3333, 21, 0, 9) Equation (3)
7 : $CS_{01} = (G_{01} - G_{02}) \otimes$ INorm(0.5, 0.02, 1.5, 0.25) Equation (5)
8 : $CS_{02} = (G_{01} - G_{03}) \otimes$ INorm(0.5, 0.02, 1.5, 0.25) Equation (5)
9 : $CS_{901} = (G_{901} - G_{902}) \otimes$ INorm(0.5, 0.02, 1.5, 0.25) Equation (5)
10 : $CS_{902} = (G_{901} - G_{903}) \otimes$ INorm(0.5, 0.02, 1.5, 0.25) Equation (5)
11 : $MFR = (CS_{01} + CS_{02} + CS_{901} + CS_{902}) \otimes$ INorm(0.5, 0.02, 1.5, 0.25) Equation (5)

---

*2.3. Noise Elimination*

The amount of noise elimination [28,29] will directly affect the difficulty of extracting the real target. From the analysis of Figure 7, it is found that fusion results still exist in a large number of non-uniform sea wave noise as well as cloud noise in both directions, how to suppress the noise becomes the research focus of this section. Further observation revealed that the targets had significant texture features in both horizontal and vertical directions after the multi-scale feature fusion result image, such as T1 in Figure 7 T1_H T1_V, T2_H T2_V, T3_H T3_V. Since sea waves have periodic flow and only have significant unidirectional texture features, such as Figure 7 N2_H N2_V or multi-direction is not significant texture features, such as Figure 7 N1_H N1_V. This leads to three conclusions:

Conclusion 1: It is a false target that does not have significant texture features in both horizontal and vertical directions.

Conclusion 2: It is a false target that has significant texture characteristics in a single horizontal or vertical direction.

Conclusion 3: It is the real target that has significant texture features in both the horizontal and vertical directions.

According to the above three conclusions, a CC calculation method is proposed, as shown in Equation (6), where *CCS* represents the CC response value and *CCF* is the CC factor, which determines the response degree of the final result. *H* and *V* are the extracted horizontal and vertical texture information respectively, and *Hmean* and *Vmean* are the averages of horizontal and vertical texture information, respectively. Details of proof process can be found in Appendix A.1.

$$CCS = \begin{cases} e^{(-CCF \times (\frac{|H-V|}{\min(H,V)}))}, & Others \\ 0, & H \leq Hmean \text{ or } V \leq Vmean \end{cases} \tag{6}$$

The final multi-scale feature fusion result obtained by CC calculation is shown in Figure 8. It can be found that sea waves and clouds are effectively eliminated. In addition, we calculated the noise numbers of Figure 7(a1,a2) to be 65 and 38, respectively, and the noise number of the CC processing result was reduced to 29, and the noise numbers of Figure 7(b1,b2) were, respectively, 105 and 81, the noise number of the CC processing result is reduced to 33.

---

**Algorithm 2:** Cross-correlation calculation

---

**Input:**
Horizontal texture feature map *H*, Vertical texture feature map *V*, correlation factor $\delta$.
**Output:**
Cross-correlation map *CC*
1: The size of image is $W \times H$
2: **for** $x$ = 1: W **do**
3:   **for** $y$ = 1: H **do**
4:     **if** $H(x,y) \leq \frac{1}{W*H} \sum_{i=1}^{W} \sum_{j=1}^{H} H(i,j) \; \Big|\Big| \; V(x,y) \leq \frac{1}{W*H} \sum_{i=1}^{W} \sum_{j=1}^{H} V(i,j)$ **do**
5:       CC = 0
6:     **else**
7:       CC = $\exp(-CF * (H - V/\min(H,V)))$
8:     **end if**
9:   **end for**
10: **end for**

---

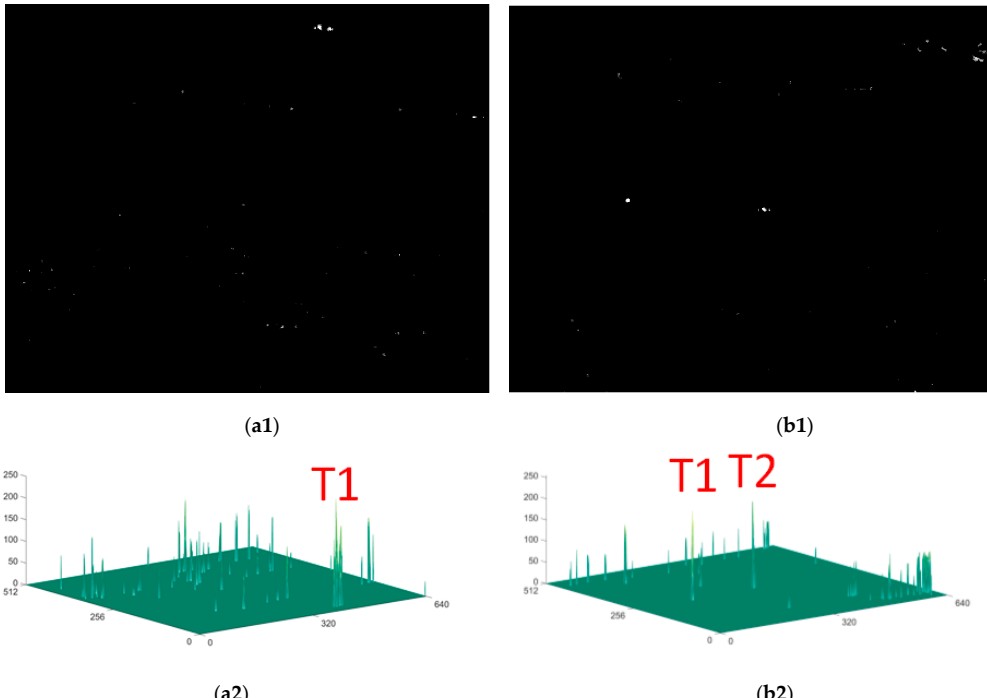

(a1)                                          (b1)

(a2)                                          (b2)

**Figure 8.** CC calculation results of Horizontal and vertical IsD multi-scale feature fusion image (Algorithm 2). (**a1**) CC result of image A. (**a2**) Three dim representation of CC result of image A. (**b1**) CC result of image B. (**b2**) Three dim representation of CC result of image B.

### 2.4. Target Enhancement

Target enhancement [30,31] can improve the generalization ability of the final segmentation model, it will make it easier to extract the target from the infrared image. The CC performance eliminates most of the maritime noise and sky background interference in the multi-scale feature fusion result, however, there are still some special image data focused on sea wave noise or sky background and the target has a similar CC result. In order to enable the algorithm to accurately segment the real target, this section studies from the perspective of target enhancement, a method based on peak significance remeasurement is proposed. Physically, there is often a radiation center inside a small target, and the energy decays as the radiation center extends outward, and attenuation degree is basically the same, so the infrared image shows a small number of extreme points around the target. When the sea waves with periodicity and continuity flow to the highest point, the reflection

of solar radiation energy will produce more peak points, the cloud layer also has more peak points due to strong edge texture characteristics.

Based on the above assumptions, the CC result is mapped to the original image as a mask to obtain an eight-direction local maximum, retained the largest peak point in the same connected domain (for the convenience of observation, carries out the morphological expansion operation). The result is shown in Figure 9. By analyzing the difference between the noise with the strongest interference in Figure 9 (such as N1) and the target (such as T1 T2), it is confirmed that the number of peak points of T1 and T2 in a certain range is far less than the number of peak points around N1. In addition, the CC result shows that the target is more significant than the clutter interference. Therefore, the following two conclusions are obtained:

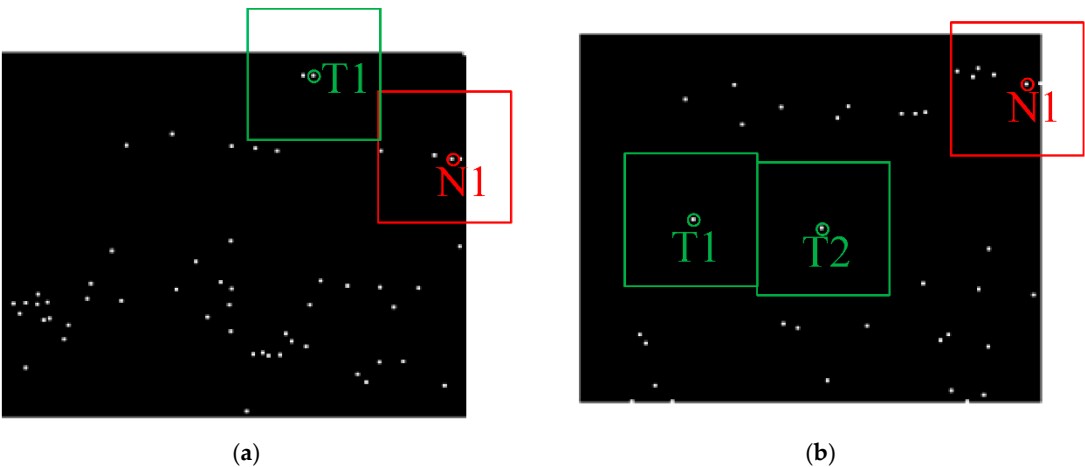

(a)                                            (b)

**Figure 9.** Local maximum detection results. (**a**) Local maximum result of image A. (**b**) Local maximum result of image B.

Conclusion 1: the real target is more likely to have less peak aggregation degree, otherwise it is the false target.

Conclusion 2: if the significance of the center peak point is stronger than the neighborhood peak point, the significance of the center peak point will be enhanced, on the contrary, weaken the intensity of the central peak point.

Conclusion 1 is achieved by weighting the number of peak aggregations in a certain range. The mathematical expression is shown in Equation (7). Among $\omega_A$ is the weighting factor for the number of peaks clustered at point A. $Nr_A$ represents the total number of peak points covered by the area radius with A as the center, and $N$ represents the total number of peak points in the entire image. Details of the proof process can be found in Appendix A.2.

$$\omega_A = 1 - \frac{Nr_A}{N} \tag{7}$$

According to Equation (7), The final detection effect is directly related to the selection of neighborhood radius $r_A$. different neighborhood radius *scales* are selected to carry out multiple groups of experiments, and the experimental results are shown in Figure 10. Where (a1) and (b1) count the number of target peak aggregations and the number of background peak aggregations under different neighborhood radius *scales*, (a2) and (b2) are the weighted coefficients of the number of target peak aggregations calculated by the statistical number of target/background peak aggregations according to Equation (7). It can be seen from Figure 10(a2,b2) that when a parameter *scale* is 81, the target has the largest weighting coefficient of peak aggregation number compared with other scale windows. In other words, the target enhancement effect is best when the parameter *scale* is 81.

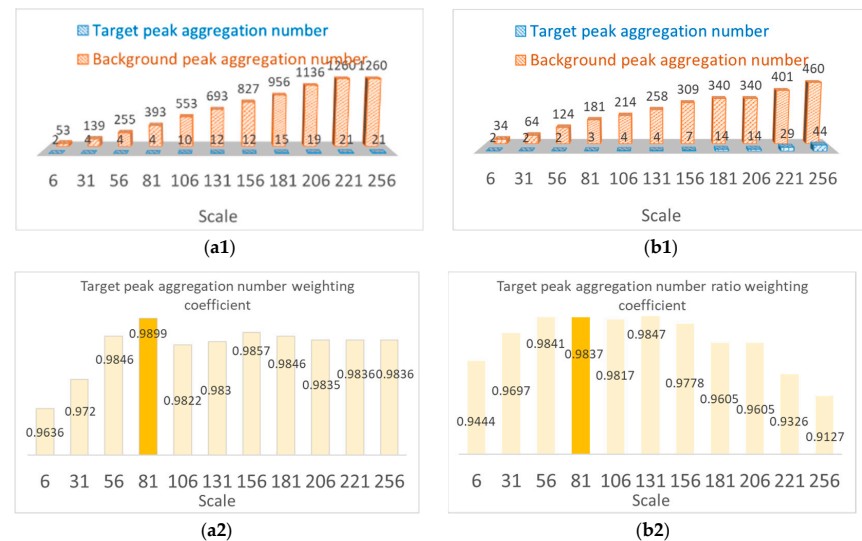

**Figure 10.** The number of target/background peak aggregation and the weighting coefficient of target peak aggregation in different *scale* windows. (**a1**) Peak aggregation number of image A. (**a2**) Weighted coefficient of target in image A. (**b1**) Peak aggregation number of image B. (**b2**) Weighted coefficient of target in image B.

Conclusion 2 is achieved by the significant difference coefficient between the center peak point and the neighborhood peak point, The mathematical expression is shown in Equation (8). $\psi_A$ is the significant difference coefficient between peak A and neighborhood peak, $S_A$ and $S_i$ are the significance values of the central peak point A and neighborhood peak point i. Details of the proof process can be found in Appendix A.2.

$$\psi_A = \frac{\sum_{i=B}^{C}(S_A - S_i)}{S_A} \tag{8}$$

The final peak significance re-measurement result of point A can be based on Equation (9). $R_A$ is the re-measurement significance result of point A. The final target's $\omega_A$, $\psi_A$ and $S_A$ values must be greater than the noise's $\omega_A$, $\psi_A$ and $S_A$ values, the calculated result target's $R_A$ value is much larger than the noise. The final result after the peak significance measurement is shown in Figure 11. We calculated the significance of target T1 in Figure 8a is 180, which is 203 after significance re-measurement. In Figure 8b, the significant values of target T1 and T2 are 168 and 150. After significance re-measurement, the significant values are 220 and 210. It can be found that the re-measurement of peak saliency achieves the improvement of target saliency and achieves the effect of target enhancement compared with Figure 8.

$$R_A = \omega_A * \psi_A * S_A + S_A \tag{9}$$

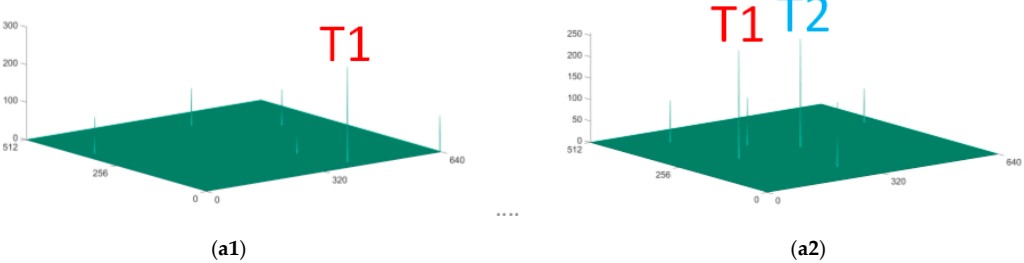

**Figure 11.** Peak significance re-measurement results of images A and B (Algorithm 3). (**a1**) Re-measurement three dim result of image A. (**a2**) Re-measurement three dim result of image B.

---

**Algorithm 3:** Peak saliency re-measurement and binary segmentation

---

**Input:**

Total number of peak points **N**. Window radius r. Horizontal and vertical coordinates of peak point **LX**, **LY**. Cross-correlation map **S**.

**Output:**

Peak saliency re-measurement result map **R**.

**Initialize:**

$Nr_A = 0$, $SD = 0$;

1: **for** $n = 1$: N **do**

2:  **for** $i = LX(n) - r$: $LX(n) + r$ **do**

3:   **for** $j = LY(n) - r$: $LY(n) + r$ **do**

4:    **if** $S(i,j) > 0$ **do**

5:      $Nr_A = Nr_A + 1$

6:      $\psi(n) = SD + \frac{S(LX(n),\ LY(n)) - S(i,j)}{S(LX(n),\ LY(n))}$

7:    **end if**

8:   **end for**

9:  **end for**

10: $\omega(n) = 1 - Nr_A / N$

11: **end for**

12: **for** $n = 1$: N **do**

13: $R(n) = \overline{\omega}(n) \times \psi(n) \times S(LX(n),\ LY(n)) + S(LX(n),\ LY(n))$

14: **end for**

---

### 2.5. Target Segmentation

Accurate target segmentation [32,33] is the main step to detect the real target. In order to make the segmentation results have higher stability and lower time-consumption, we perform the binarization segmentation processing on the peak significance remeasurement result, and the calculation is shown in Equation (10), where *avg* and *std* are maximum value point mean and standard deviation, φ is an adjustable factor to adjust the intensity of thresholding after binarization segmentation. in order to maintain the original area of the target as much as possible, the segmented result is applied as the seed point to growing the region of eight connected domains. The final result after binarization segmentation is shown in Figure 12.

$$Thresh = avg + \varphi \times std \tag{10}$$

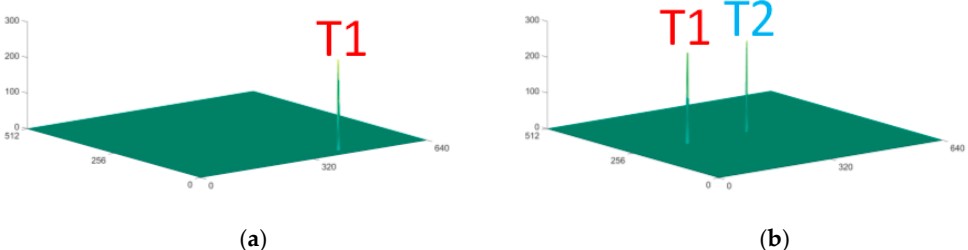

(a)                                                                 (b)

**Figure 12.** Binary segmentation results of images A and B. (**a**) Image A segmentation result. (**b**) Image B segmentation result.

### 3. Results and Discussion

In this Section, in order to evaluate the performance index more accurately of the proposed algorithm. In Section 3.1, validation data sets are introduced and compared with five kinds of target detection algorithms, which are Partial Sum of Tensor Nuclear Norm, Wavelet Transform, Anti-jitter Spatiotemporal Saliency Generation with Parallel Binarization, Visual Attention and Pipeline Filtering Model, Local Steering Kernel. At the same time, the detailed parameter settings of five detection methods are provided. In Section 3.2, qualitative results are presented to demonstrate the effectiveness of our method and demonstrate the effectiveness of the proposed strategy and comparison algorithm in

detection results. Section 3.3 introduced some evaluation indicators, signal-to-clutter ratio gain, background suppression factor, time complexity, and receiver operating characteristic curve (ROC) are analyzed.

### 3.1. Datasets Introduction

With the help of the long-wave infrared (Wavelength: 8–12 um Resolution: 640 × 512 brand: FLIR) search and rescue system developed by the laboratory, a series of image sequences (Digital quantization: eight-bit) of SWWBC in a variety of experimental environments were captured. A representative image of each sequence is shown in the first line of Figure 13 Data set A consists of 350 infrared maritime images with low target intensity, weak sea waves, and thick cloud background. Data set B is composed of 250 infrared maritime images with low sky background intensity and strong sea wave background, and the gray distribution of the target is uneven due to the influence of sunlight. Data set C contains 450 images of only sea waves, and the sea waves show uneven brightness distribution. Data set D are composed of 50 images with a smooth background of sea wave and sky. Data set E and data set F consist of 100 and 150 infrared maritime images with different numbers of independent island interference, respectively. The two types of data sets are captured in different fields of view, data set E contains a target with weak contrast. Data set G and data set H contain 100 and 150 infrared images with connected island interference, respectively, data set H is closer to the infrared detector, and the size of the target is larger.

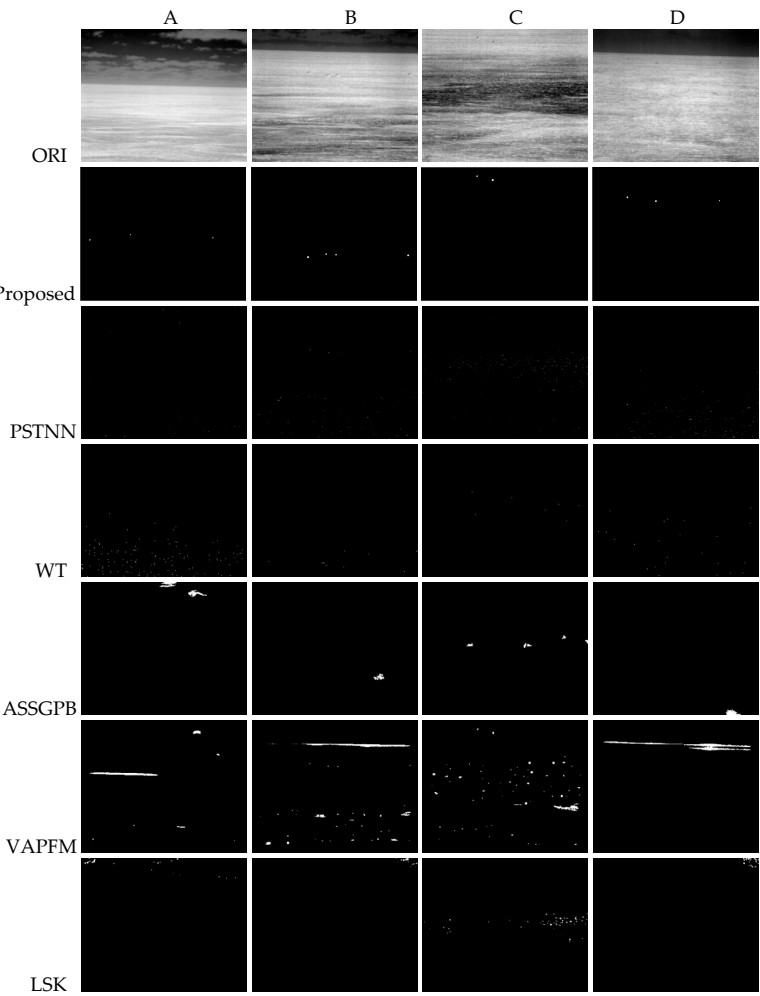

**Figure 13.** *Cont.*

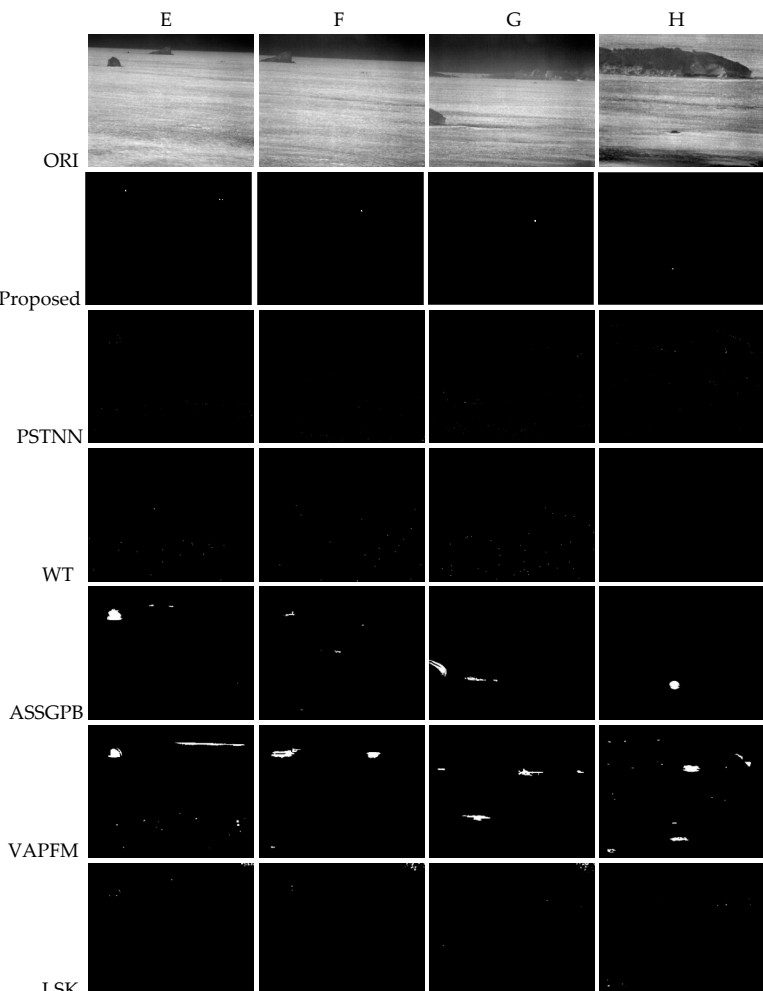

**Figure 13.** Detection results of different algorithms.

### 3.2. Qualitative Comparisons

Considering that the inversion problem has had an extensive impact on image analysis and computer vision in recent years and has achieved remarkable target detection results. The partial sum of the tensor nuclear norm can extract real infrared small targets from complex backgrounds. The wavelet transform method with multi-scale and multi-directional analysis ability can accurately extract weak and small targets. Considering that the anti-jitter spatiotemporal saliency generation with the parallel binarization method can detect targets in the backlight marine environment. Combining single-frame visual attention mechanism detection of suspected targets and multi-frame pipeline filtering to determine the real target is a stable small target detection scheme in sea waves conditions. According to the local area of the strong wind and wave backlit image, the target intensity is low, and the background intensity is high. Therefore, the target local patch constitutes a strong contrast. Based on the local steering kernel local descriptor of the human visual system, the neighborhood is coded with contrast features, so as to achieve accurate weak and small target detection. In order to analyze the performance advantages of each algorithm more justly, the detailed parameter settings of five detection methods are listed in Table 1. Captured data sets cover the common challenges of infrared maritime target detection in SWWBC. The excellent detection performance of these data sets can verify the effectiveness of the algorithm.

**Table 1.** Parameter setting of state-of-art methods.

| Method | Abbreviation | Parameter Setting |
|---|---|---|
| Partial Sum of Tensor Nuclear Norm [10] | PSTNN | Patch size: $40 \times 40$. sliding step: 10. $\varepsilon = 10^{-7}$. $\lambda = 0.6/\sqrt{\max(n1, n2) * n3}$. |
| Wavelet Transform [8] | WT | Wavelet base: haar. A slide $6 \times 6$ pixels window. The threshold value of judgment of horizontal or oblique is 20. |
| Anti-jitter Spatiotemporal Saliency Generation with Parallel Binarization [21] | ASSGPB | background excitation factor: $\sigma_B = 0.5 + 0.15 * width$. foreground inhibition factor: $\sigma_F = 2.7 - 6.25e^{-4} * width$. |
| Visual Attention and Pipeline Filtering Model [20] | VAPFM | Classification threshold: 0.016. Block Size: $32 \times 32$. |
| Local Steering Kernel [5] | LSK | global smoothing parameter h = 0.2. overlap size t = 2. the number of neighboring patches N = 8. |

Figure 13 shows the typical images captured and the corresponding results generated by different experimental algorithms. From the results of data set A, the proposed method can accurately detect the target and completely eliminate the background clutter interference. The WT method can detect some targets whereas it also produces a lot of false targets. Other methods cannot detect the real target. From the results of data set B, the proposed method, VAPFM, and PSTNN method can successfully detect four sea targets with bright spots, but VAPFM and PSTNN detected a large number of false alarms, other methods do not have the ability of accurate detection. From the results of data set C, the proposed method and VAPFM method can successfully detect the maritime targets without sea-sky background, but the number of false alarms in VAPFM is unacceptable, other methods cannot be accurately extracted due to the influence of target intensity and uneven sea wave background distribution. From the results of data set D, the proposed method has a good detection effect when the target contrast is weak. Other methods cannot effectively detect small targets with weak contrast. From the results of data set E and F, the proposed method can eliminate the interference of independent islands. Other methods are missed due to the intensity and size of the target and the influence of gradient and shape features of the island. From the results of data set G, the proposed method and WT method can effectively suppress the contiguous islands. However, the WT method is sensitive to gradient, resulting in a lot of false detection. Other methods are missed due to the intensity and size of the target and the influence of gradient and shape features of the island. From the results of data set H, the proposed method, ASSGPB and VAPFM methods can accurately detect large targets, while other methods are trapped by the interference of sea wave intensity with uneven distribution and connected islands cannot determine the real target.

The main defects of the comparison method can be summarized as follows. Compared with the regular environmental wave background, the PSTNN method is difficult to achieve a low rank in the backlit ocean wave environment, which increases the difficulty of the target sparsity constraint, which makes it difficult to accurately distinguish the target and the background. Experiments show that the WT method is sensitive to gradient, so it will produce a higher false alarm rate when dealing with strong wind-wave backlight conditions. In strong wind-wave backlight conditions, the gray distribution of the target will be changed by the interference of the wave background, so the assumption of temporal and spatial consistency in ASSGPB is no longer applicable, resulting in the inability to continuously detect weak and small targets. Although VAPFM can overcome flowing waves, it cannot suppress the interference of islands with stable strength, resulting in a higher false alarm rate, and weak targets are easily eliminated. Experiments show that LSK

local descriptor will produce a stronger response value only when there is a uniform and low-intensity background around the target, it does not have the ability to characterize weak and small targets in strong wind-wave backlight conditions.

### 3.3. Quantitative Comparisons

We select the signal-to-clutter ratio gain and the background suppression factor as evaluation indices. The *SCRG* value and the *BSF* value are defined as:

$$SCRG = \frac{(S/C)_{out}}{(S/C)_{in}} \tag{11}$$

$$BSF = \frac{C_{in}}{C_{out}} \tag{12}$$

where *S* and *C* are the average target intensity and clutter standard deviation respectively. The $(\cdot)in$ and $(\cdot)out$ are the original image and the result of the method. The signal-to-clutter ratio gain index measures the magnification of the target relative to the backgrounds before and after processing. The background suppression factor represents the suppression effect of backgrounds without any information about the target. Experimental results of these methods with the index are shown in Table 2. The highest value of each evaluation index in each column is marked red, and the second-highest one is marked blue. it can be analyzed that the proposed method achieves the highest values of *SCRG* and *BSF* in A-G. The proposed method can improve *SCRG* to some extent, it means the detected target is more prominent than the backgrounds while achieving a better suppression effect on the background. That is to say, the proposed method outperforms the compared methods in both target enhancement and background suppression from the angle of numerical indicator values for these data sets. For H, the proposed method has the second-highest *SCRG* and *BSF*, which is smaller than the ASSGPB method because the method is more suitable for detecting large-size targets. However, the difference between the proposed method and the comparison methods is small.

**Table 2.** Evaluation indices comparison of SCRG, BSF.

| Method | Evaluation Indices | A | B | C | D | E | F | G | H |
|---|---|---|---|---|---|---|---|---|---|
| Proposed | SCRG | 21.9 | 18.2 | 35.4 | 34.2 | 36.1 | 33.5 | 35.5 | 27.2 |
| | BSF | 6.2 | 7.8 | 10.8 | 10.7 | 9.47 | 9.6 | 7.9 | 7.5 |
| PSTNN | SCRG | 9.5 | 13.1 | 11.3 | 10.8 | 8.5 | 8.2 | 5.3 | 4.8 |
| | BSF | 3.5 | 8.2 | 3.1 | 4.2 | 3.2 | 3.4 | 3.1 | 2.5 |
| WT | SCRG | 13.1 | 9.8 | 14.4 | 17.6 | 15.1 | 13.1 | 14.8 | 12.5 |
| | BSF | 5.4 | 3.3 | 6.6 | 6.7 | 4.3 | 4.2 | 4.7 | 4.8 |
| ASSGPB | SCRG | 1.9 | 2.1 | 5.0 | 6.6 | 4.5 | 24.5 | 3.6 | 36.8 |
| | BSF | 1.8 | 2.2 | 4.1 | 6.0 | 3.1 | 7.4 | 4.1 | 9.0 |
| VAPFM | SCRG | 8.9 | 7.8 | 18.5 | 18.7 | 16.6 | 16.2 | 15.9 | 15.6 |
| | BSF | 3.4 | 2.3 | 8.1 | 8.8 | 6.7 | 6.1 | 5.8 | 4.1 |
| LSK | SCRG | 3.2 | 5.8 | 4.3 | 4.8 | 3.3 | 5.2 | 4.3 | 3.8 |
| | BSF | 1.3 | 1.6 | 2.1 | 4.1 | 2.2 | 2.4 | 3.1 | 2.5 |

Table 3 shows the algorithm complexity. Suppose the image size is and *m*, *n* are the rows and columns of the image. The computational cost of WT is O ($mn \log mn$), the main time-consuming part is wavelet decomposition. The computational cost of the proposed method and LSK is O ($l^2 \log l^2$), where the $l^2$ is the size of the optimal window. Considering the image size, the final cost of the proposed method and LSK is O ($l^2 \log l^2 mn$). For VAPFM and ASSGPB, it is obvious that the major time-consuming part is calculating the

saliency map pixel by pixel. A sliding window of size is needed for computing the saliency value of the central pixel. Thus, $k^2$ times mathematical operation per pixel is required, namely, in a single scale, The time complexity of $i$ scale is O $(ik^2mn)$. As the ASSGPB is a multi-frame method, the final complexity is O $(sik^2mn)$, S is the number of images in the pipeline. For the PSTNN methods, the dominant factor is singular value decomposition (SVD), the size of the patch-tensor is $n1 \times n2 \times n3$, the dominant factor of the complexity cost in calculating the SVD and FFT, the final computation cost of PSTNN model is O $(n1n2n3logn1n2)$.

**Table 3.** Comparison of computational complexity of six methods.

| Method | Proposed | PSTNN | WT |
|---|---|---|---|
| Complexity | O $\left(mnl^2 \log l^2\right)$ | O $\left(n1n2n3logn1n2\right)$ | O $\left(mn \log mn\right)$ |
| Method | ASSGPB | VAPFM | LSK |
| Complexity | O $\left(sik^2mn\right)$ | O $\left(ik^2mn\right)$ | O $\left(mnl^2 \log l^2\right)$ |

To further demonstrate the advantages of the developed method, we provide the ROC curves of the test sequences in Figure 14. ROC curve is usually used to assess detection performance and represents the varying relationship of detection rate and false alarm rate. The horizontal coordinate of the ROC curve is false alarm rate and the vertical coordinate is detection rate, mathematical expression is as follows:

$$DR = \frac{DT}{AT} \tag{13}$$

$$FAR = \frac{FD}{FD + DT} \tag{14}$$

where $AT$ denotes the total number of real target detections in the image sequence, $DT$ denotes the total number of detected real targets and $FD$ is the total number of false targets, which are the residual clutters. Furthermore, if there is an overlapping area between the detected target and the ground truth target, then the detected target will be taken as the actual target. If the distance between two detected targets is within a certain range (two pixels), the two targets will be regarded as one target.

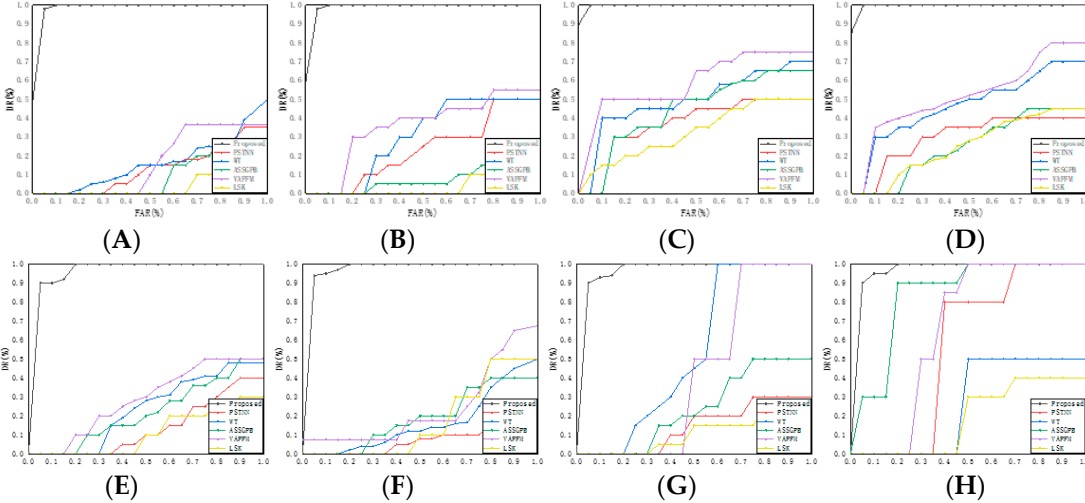

**Figure 14.** ROC curves of the six methods on datasets (**A–H**), respectively.

The results show that the detection rate of the proposed algorithm can reach 100% without island and cloud background interference (CD), and the maximum false alarm

rate is not more than 0.05. When there is thick cloud interference (AB), the detection rate can also reach more than 0.95, and the maximum false alarm rate is not more than 0.08, which is the optimal result. When there is island interference (EFGH), the detection rate can reach more than 0.92, and the maximum false alarm rate is less than 0.16. In addition, it can be observed that when the false alarm rate is not more than 0.05, the detection rate of the proposed method is higher than other comparison methods. In other words, the proposed method almost achieves the highest detection rate and the lowest false alarm rate. It means that the proposed method is superior to other latest methods.

## 4. Conclusions

To achieve the purpose of accurate detection of distress targets in SWWBC, a small target detection algorithm in SWWBC is proposed. Our main work is divided into five parts, which are: background suppression using IGD method; Multi-scale Gabor filter; the IsD operator; iterative normalization operator; and multi-scale fusion are applied to realize feature extraction.

More specifically, the main contributions of the proposed target detection method are as follows.

(1) We propose an IGD preprocessing method which is similar to the pixel distribution of the target block image to suppress the uniform wave and sky background intensity and achieve the purpose of background suppression.

(2) We apply a multi-scale Gabor filter to extract horizontal and vertical directional texture features, combine the proposed IsD operation and iterative normalization operator to highlight the target, and suppress noise interference.

(3) According to the fact that the target has more significant texture intensity than the noise in the horizontal and vertical directions, we propose a CC method to remove the noise.

(4) According to the dispersion of the number of extreme points and the significance of the intensity of the small target compared with the sea wave and sky noise, we propose a peak significance re-measurement method to make the target prominent and combine with a binary method to achieve accurate target segmentation.

The performance of infrared target detection can be evaluated from five aspects: image quality, time complexity, detection rate, false alarm rate, and stability. The final experimental results show that the method has a higher background suppression factor and signal-to-clutter ratio gain, indicating that the processed background is suppressed, and the target is more prominent than the background, which is beneficial to the final target detection. The proposed method has low time complexity and has the potential of parallel processing, which can meet the real-time requirements. According to the ROC curve obtained from the final statistics, it can be found that compared with other state-of-art methods, the proposed algorithm can achieve the maximum detection rate while reducing the false alarm rate as much as possible, and the area under the ROC curve of different data sets is similar, so the proposed algorithm has excellent stability.

**Author Contributions:** D.M. provided the initial idea and wrote the manuscript; D.M. designed and performed the research; L.D. and W.X. helped in the discussion and partially financed the research. All authors have read and agreed to the published version of the manuscript.

**Funding:** This paper was supported in part by the Fundamental Research Funds for the Central Universities of China under Grant 3132019340 and 3132019200. This paper was supported in part by high-tech ship research project from the ministry of industry and information technology of the people's republic of China under Grant MC-201902-C01.

**Acknowledgments:** The authors would thank the published code of Zhang's model (PSTNN), Wang's model (ASSGPB), Li's model (LSK), Dong's model (VAPFM) and Wei's model (WT) for comparison.

**Conflicts of Interest:** The authors declare no conflict of interest.

**Appendix A.**

As follows, we will prove the conclusions presented in Section 2.3 noise elimination and Section 2.4 target enhancement.

*Appendix A.1. A CC Calculation Method*

Proof conclusion 1: For Equation (6), when the horizontal and vertical directions do not have obvious texture features, the significant value of these "unobvious" positions is generally less than the average value, when $H <= Hmean$ and $V <= Vmean$, $CCS = 0$. Corresponding pixel position is judged as a false target.

Proof conclusion 2: For Equation (6), when a single direction in horizontal or vertical has obvious texture characteristics, that is, the value of $H$ and $V$ is different, the value of $|H - V|$ is bigger, and the value of $1/\min(H, V)$ is also greater. So $|H - V|*(1/\min(H,V))$ has a larger value, so $\exp(-CF * (H - V/\min(H, V))) -> 0$, that is, $CCS -> 0$, corresponding pixel position is judged to be a false target.

Proof conclusion 3: For Equation (6), when both the horizontal and vertical directions have obvious texture features, that is, the values of $H$ and $V$ are higher than their respective mean values and the difference between the values of $H$ and $V$ is small, then $|H - V| -> 0$, $1/\min(H,V)$ is smaller, $|H - V|*(1/\min(H,V)) -> 0$, $\exp(-CF * (H - V/\min(H, V))) -> 1$, that is, $CCS -> 1$. Corresponding pixel position is judged as the real target.

*Appendix A.2. Peak Significance Re-Measurement Method*

Proof conclusion 1: Figure A1 shows a diagram for calculating the weighting factor of the number of peak clusters, where $Nr_A = A + B + C$, $N = Nr_A + D + E + F$. If point A is a noise point, $Nr_A$ value is larger, the weighting factor $\omega_A$ is smaller. On the contrary point, A is the target, $Nr_A$ value is smaller, the weighting factor $\omega_A$ is larger.

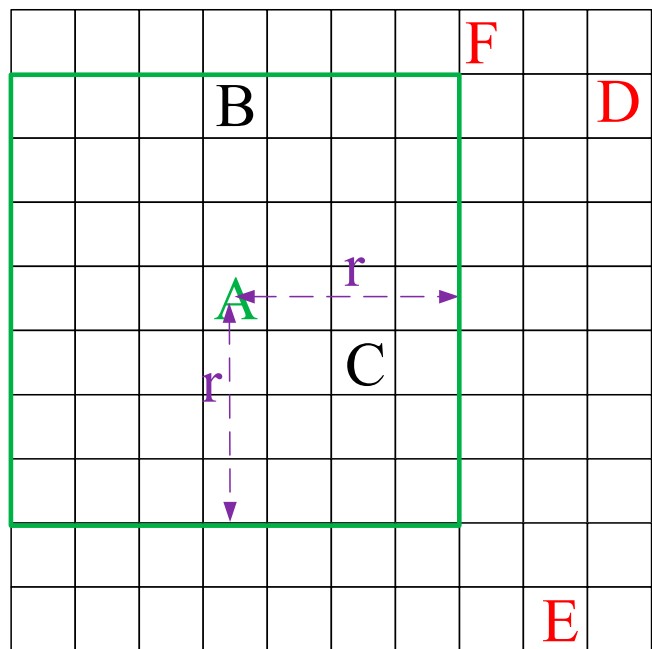

**Figure A1.** Schematic diagram of weighted coefficient calculation of peak aggregation number.

Proof conclusion 2: If point A is a noise point and the value of $S_A$ is close to or smaller than $S_i$, then the value of the weighting coefficient $\psi_A$ is small or negative. On the contrary, if point A is the target and the value of $S_A$ is much larger than $S_i$, the result of the weighting coefficient $\psi_A$ value is greater.

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
