# Peer review of "A Method of Infrared Small Target Detection in Strong Wind Wave Backlight Conditions"

_remotesensing, doi:10.3390/rs13204189_

Round 1
Reviewer 1 Report
The paper is well written.
The topics are illustrated in a clear and complete manner. The solutions found have been investigated correctly. The comparison with the other methods present in the literature was carried out rigorously.
Just some a few more references could be cited.
Author Response
Thank you very much for the reviewers' approval of the latest work reported. I agree with this opinion. Some useful references related to the content of the reported work are cited.
- Kocak, H.; and Altintas, H. Evaluation of maritime accident reports of main search and rescue coordination centre between 2001 and 2012. International maritime health, 2021, 72(3), 163-171. doi:10.5603/IMH.2021.0032.
- Zhou, F.; Chen, H.; and Zhang, P. Performance Evaluation of Maritime Search and Rescue Missions Using Automatic Identification System Data. Journal of Navigation, 2020,1-10. doi:10.1017/S0373463320000223.
- Leonard, C. L.; Deweert, M. J.; and Gradie, J. Performance of an EO/IR sensor system in marine search and rescue. Proceedings of SPIE - The International Society for Optical Engineering, USA, 2005, 122–133. doi: 10.1117/12.603909.
- He, L.; Liu, C.; and Li, J. Hyperspectral Image Spectral–Spatial-Range Gabor Filtering. IEEE Transactions on Geoscience and Remote Sensing, 2020, 58(7), 4818-4836. doi:10.1109/TGRS.2020.2967778.
- Zhang, J.; Zhou, Q. Wu, J. A Cloud Detection Method Using Convolutional Neural Network Based on Gabor Transform and Attention Mechanism with Dark Channel Subnet for Remote Sensing Image. Remote Sensing, 2020, 12(19), 3261. doi: 10.3390/rs12193261.
- Tu, B.; Li, N. Y.; Fang, L. Y.; He, D. B.; and Ghamisi, P. Hyperspectral Image Classification with Multi-Scale Feature Extraction. Remote Sensing, 2019, 11(5), 534. doi: 10.3390/rs11050534.
- Zhang, Y.; Zhang, L.; and Bai, X. Infrared and Visual Image Fusion through Infrared Feature Extraction and Visual Information Preservation. Infrared Physics & Technology, 2017, 83. doi: 10.1016/j.infrared.2017.05.007.
- Tian, T.; Pan, Z.; Tan, X. Arbitrary-Oriented Inshore Ship Detection based on Multi-Scale Feature Fusion and Contextual Pooling on Rotation Region Proposals. Remote Sensing, 2020, 12(2), 339. doi: 10.3390/rs12020339.
- Jones, J. P.; Palmer, L. A. An evaluation of the two-dimensional Gabor filter model of simple receptive fields in cat striate cortex. Journal of Neurophysiology, 1987, 58(6), 1233-1258. doi: 10.1016/0165-5728(87)90046-4.
- Da, C.; Shao, X.; and Hu, B. A Background and noise elimination method for quantitative calibration of near infrared spectra. Analytica Chimica Acta, 2004, 511(1), 37-45. doi: 10.1016/j.aca.2004.01.042.
- Meng, S.; Noise Elimination and Contour Detection Based on Innovative Target Image Contour Coding Algorithm. Shock and Vibration, 2020, 2020, 1-8. doi: 10.1155/2020/8895000.
- Chen, L.; Jiang, X.; Li, Z. Feature-Enhanced Speckle Reduction via Low-Rank and Space-Angle Continuity for Circular SAR Target Recognition. IEEE Transactions on Geoscience and Remote Sensing, 2020, 99, 1-19. doi: 10.1109/TGRS.2020.2983420.
- Bai, X.; Zhou, F.; Xue, B. Infrared dim small target enhancement using toggle contrast operator. Infrared Physics & Technology, 2012, 55, 177–182. doi: 10.1016/j.infrared.2011.12.002.
- Shang, R.; Zhang, J.; Jiao, L. Multi-scale Adaptive Feature Fusion Network for Semantic Segmentation in Remote Sensing Images. Remote Sensing, 2020, 12(5), 872. doi: 10.3390/rs12050872.
- Huang, L.; Dai, S.; and Huang, T. Infrared Small Target Segmentation with Multiscale Feature Representation. Infrared Physics & Technology, 2021, 116. doi: 10.1016/j.infrared.2021.103755.
Reviewer 2 Report
The topic is contemporary and of interest. The language needs to be revised to avoid long sentences (the first sentence in the Abstract for example) and correct mixed tenses, verb conjugation, plural vs singular, punctuation marks (comma vs full stop) and other similar issues.
Two of the 18 references are self-reference, which is not a big percentage, but puts into question the novelty of the reported work.
It is not clear why the particular five methods are selected for comparative analysis, nor what are their main deficiencies as compared to the proposed method.
Some conclusions are overlapping and their proofs are not necessarily following mathematical derivation. Perhaps the authors would consider moving the proofs into an appendix or reformulating them into a more meaningful, contributing items.
Author Response
Thank you again for your positive comments.The detailed response is shown in the PDF.

Reviewer 3 Report
This paper deals with the detection of small objects at sea in adverse conditions of wind and waves using thermal images obtained with an infrared camera. It is true that the treatment and processing of infrared images is tremendously complex due to the lack of definition of the edges of the objects and if we add to this the waves that move the objects and add noise, it justifies its complexity.
The paper addresses this problem by using an object segmentation and extraction algorithm based on five steps: background suppression, feature extraction, noise removal, object enhancement and object segmentation.
The structure of the paper is correct, a correct description of the problem is made, the state of the art with the different algorithms proposed by other authors is addressed, a description of the proposed method is made, the different steps of the algorithm are analyzed, the algorithm is applied to a set of reference images containing different cases or possibilities and concludes with an analysis of results.
The inclusion of pseudocode to expose the different filters and procedures that compose the algorithm that clarify its usefulness and application is appreciated.
It would be interesting to review the placement of the images in the text and try to improve as much as possible their resolution to better appreciate the results.
In general, I do not find significant elements not to consider this paper publishable and extend my congratulations to the authors for the work done.
Author Response
Thank you very much for the reviewers' approval of the latest work reported. On this basis, we have improved the text size in the figure, grammar and other similar problems.
Round 2
Reviewer 2 Report
The authors have addressed the concerns expressed in the comments.